# Opioids and Sepsis: Elucidating the Role of the Microbiome and microRNA-146

**DOI:** 10.3390/ijms23031097

**Published:** 2022-01-20

**Authors:** Yaa Abu, Nicolas Vitari, Yan Yan, Sabita Roy

**Affiliations:** 1Medical Scientist Training Program, Miller School of Medicine, University of Miami, Miami, FL 33136, USA; yfa8@med.miami.edu; 2Department of Microbiology and Immunology, Miller School of Medicine, University of Miami, Miami, FL 33136, USA; nav54@miami.edu; 3Department of Surgery, Miller School of Medicine, University of Miami, Miami, FL 33136, USA; yxy801@med.miami.edu

**Keywords:** opioids, sepsis, miRNA, miR-146, endotoxin, inflammation, microbiome

## Abstract

Sepsis has recently been defined as life-threatening organ dysfunction caused by the dysregulated host response to an ongoing or suspected infection. To date, sepsis continues to be a leading cause of morbidity and mortality amongst hospitalized patients. Many risk factors contribute to development of sepsis, including pain-relieving drugs like opioids, which are frequently prescribed post-operatively. In light of the opioid crisis, understanding the interactions between opioid use and the development of sepsis has become extremely relevant, as opioid use is associated with increased risk of infection. Given that the intestinal tract is a major site of origin of sepsis-causing microbes, there has been an increasing focus on how alterations in the gut microbiome may predispose towards sepsis and mediate immune dysregulation. MicroRNAs, in particular, have emerged as key modulators of the inflammatory response during sepsis by tempering the immune response, thereby mediating the interaction between host and microbiome. In this review, we elucidate contributing roles of microRNA 146 in modulating sepsis pathogenesis and end with a discussion of therapeutic targeting of the gut microbiome in controlling immune dysregulation in sepsis.

## 1. Introduction

Sepsis has recently been defined as life-threatening organ dysfunction caused by the dysregulated host response to an ongoing or suspected infection (Figure 1) [1]. To date, sepsis continues to be a leading cause of morbidity and mortality amongst hospitalized patients [2,3]. The mainstays of therapy—antibiotics and advanced life-support care—have not significantly changed the case fatality rate for patients with sepsis, which has remained between 30% and 50% over the past three decades [4,5,6,7]. Additionally, several longitudinal studies have shown an increased 6- and 12-month mortality rate in up to 20% of survivors of sepsis and septic shock, corresponding with a significantly lower health-related quality of life [4,5,6,7]. Moreover, the economic cost from sepsis, partly due to prolonged ICU stays and increased mortality rates, accounted for more than $20 billion of total US hospital costs in 2011 and are associated with a decrease in gross national income [1,6].

Research has thus focused on understanding the etiology of immune dysregulation underlying sepsis, which is now understood to be marked by simultaneous inflammation and immune suppression. In the pathogenesis of septic shock, the host inflammatory response to bacterial invasion plays a key role in determining whether sepsis may occur. Bacterial recognition and signaling by the innate immune system normally function to coordinate an appropriate immune response dedicated towards controlling the infectious process. However, in sepsis, the excessive response of the innate immune system following bacterial invasion is what results in an hyper-inflammatory cytokine storm, multiple organ failure, and death [3,8,9]. Multiple reports have shown that Toll-like receptor (TLR) systems play a key role in mediating systemic response to pathogens during sepsis. TLR signaling, in response to bacterial endotoxins, leads to downstream activation of nuclear factor kappa B (NF-κB), which mediates the transcription of pro-inflammatory mediators in the nucleus [10,11,12]. However in sepsis, dysregulated NF-κB activity initially results in a cytokine storm, which can progress to multiple organ failure [3,11,13]. Subsequent immunosuppression due to immune exhaustion and epigenetic reprogramming of innate and adaptive immune cells results in susceptibility to secondary infections [3,11,13]. Despite some appreciation of the underlying immune dysfunction in the pathogenesis of sepsis, little progress has been made in improving sepsis-related morbidity and mortality.

Furthermore, there is a growing importance of understanding the interactions between opioid use and the development of sepsis, given the widespread use and misuse of opioids. Sepsis is an important cause of morbidity and mortality in patients with opioid use disorders (OUDs). Opioid overdoses significantly contribute to sepsis hospitalizations, with a 34% increase in ICU admissions in the last 7 years [14,15]. The ongoing opioid epidemic has further increased the incidence of people at risk for sepsis, with an estimate of 10.1 million people aged 12 or older misusing opioids in 2019 alone [16]. In addition to opioid misuse, prescription opioids are commonly administered for pain management in hospital and outpatient settings, typically post-operatively or related to an injury. Globally, it is estimated that nearly 7.35 billion doses of opioids are administered to hospitalized patients each day, making investigations targeted at understanding the influence of opioids on sepsis and septic shock risk highly warranted [17].

The gut microbiome has emerged as a key player in regulating health and disease. There is increasing evidence that the gut microbiome plays a crucial role in the pathogenesis of sepsis, and that gut microbial dysbiosis predisposes to sepsis and increased morbidity. In this review, we will describe the role of the gut microbiota in sepsis, and interactions between sepsis and opioids. Here, we also elucidate contributing roles of microRNAs, in particular microRNA 146, in modulating sepsis pathogenesis and end with a discussion of therapeutic targeting of the gut microbiome in managing sepsis.

## 2. The Role of the Gut Microbiota in Sepsis

The human gut microbiota is a complex ecosystem comprising trillions of microorganisms (bacteria, archaea, fungi, protists, and viruses) residing within the human gastrointestinal tract along with their genes and metabolites. Recently, the gut microbiome has emerged as a key modulator of sepsis (Figure 2). These findings have been extensively described [6,18,19,20,21], and are supported by the observation that the intestinal tract is a major site of origin of sepsis-causing microbes [18].

### 2.1. Intestinal Microbial Homeostasis and Resistance to Pathogen Colonization

Under physiological conditions, intestinal microbial homeostasis promotes resistance to pathogen colonization [22]. For example, microbe-associated molecular patterns (MAMPs) such as flagellin and lipopolysaccharide (LPS) stimulate TLR4 on antigen-presenting cells, leading to epithelial cell release of the antibacterial lectin REGIIIγ; this confers protection against infections with enteropathogenic bacteria [6,23,24]. Gut microbial populations further function to regulate innate and adaptive immunity—including T cell differentiation and activation, antibody production, macrophage phagocytosis, and cytokine production—both locally and systemically. These microbes exert their functions on the immune system via contact-dependent mechanisms or via secretion of bacterial metabolites. Commensal bacteria in the phyla Bacteroides and Firmicutes are major producers of short-chain fatty acids (SCFA), which enhance intestinal epithelial barrier function, regulate gene expression in T cells, and alter the microbicidal function in macrophages to maintain homeostasis [6,20]. SCFA butyrate production by Clostridia and Faecalibacterium further promote regulatory T cell differentiation in the colon through upregulation of Foxp3 [25]; additionally, butyrate can inhibit histone deacetylation, which decreases the release of NF-κB-regulated pro-inflammatory cytokines regulating enteric and systemic inflammation [18,26]. Regulation of goblet cells and Paneth cells by SCFAs further promote mucous production and activate immune defenses required for proper maintenance of the gut epithelial layer [6]. This has further been demonstrated in murine models through acetate supplementation, which protects against *Escherichia coli* translocation [27].

The manipulation of the microbiota through antibiotics has emphasized the protective role of the microbiota in immune homeostasis, and how alterations in gut microflora may lead to disease. Gut microbial dysbiosis is thought to influence the inflammatory responses through increased gut barrier permeability, allowing for the translocation of pathobionts into systemic circulation. This has been well characterized in rodents through the use of germ-free or antibiotic-treated mice, which display increased mortality after bacterial challenge [19,23,28]. Specifically, there have been multiple reports that pretreatment of animals with antibiotics prior to infection with Pneumococcal pneumonia, *E. coli*, and *Klebsiella pneumoniae* worsens disease outcomes, with observed expansions of pathogenic clones of multi-drug-resistant bacteria into systemic circulation [20,29]. Hospitalized patients treated with antibiotics further display this phenotype; while antibiotics have been indispensable to infection management, they have consistently been shown to paradoxically increase susceptibility to nosocomial infections by altering colonization resistance [30]. Collectively, these studies highlight the protective role of the gut microbiota in maintaining immune homeostasis, with disruptions in gut microbial composition contributing to disease susceptibility.

### 2.2. Gut Microbial Disruption by Sepsis: A Bi-Directional Relationship

The intestinal microbiota has been characterized as a virtual organ due to its key functions in human physiology [31]. The composition of the intestinal microbiome is severely affected by sepsis, and alterations in the human microbiome in the course of sepsis are now considered to be another kind of organ failure [20,21]. Antibiotics are commonly used to manage critically ill septic patients. In a global study of 1265 ICUs, it was found that 75% of admitted patients received antibiotics during their hospital stay [32]. Disruptions in the gut microbiome with antibiotics to prevent sepsis and septic shock have been implicated in organ dysfunction [20]. Thus, there has been an increasing focus on understanding changes in microbiota composition during sepsis and adequately managing these imbalances to mitigate other organ failures in septic patients.

Current data suggest that septic patients may have altered gut microbiomes, implicating dysbiosis as a risk factor for sepsis through the disruption of metabolic and immune homeostasis. This has been well explored in the elderly population, who are at high risk of sepsis due to a combination of intestinal physiology deterioration, extensive medication use, and altered dietary patterns [33]. Heterogeneous patterns of intestinal microbiota in both septic and non-septic critically ill patients have been observed in a prospective observational cohort study of patients admitted to the intensive care unit [31]. These analyses have revealed that critical illness profoundly disrupts gut microbiome composition. In general, septic patients were found to have a distinct intestinal microbial community compared to those without sepsis per β-diversity evaluation [34]. These alterations included a significant decrease in microbial diversity, including loss of commensal bacteria and overgrowth of pathobionts such as *Enterococcus*, *Staphylococcus*, and *Clostridia* spp. [19]. Critically ill patients have also been demonstrated to have lower abundances of *Faecalibacterium*, *Blautia*, *Ruminococcus*, *Subdoligranulum*, and *Pseudobutyrivibrio* [31]. These bacteria have important functions in host metabolism through the production of short-chain fatty acids which maintain gut homeostasis and have anti-inflammatory properties [31]. Consistently, clinical studies have found that fecal acetate and butyrate are significantly lower in critically ill patients compared to healthy controls [35]. In murine cecal ligation and puncture (CLP) sepsis models, these differences in gut microbial composition often correspond to significant increases in mortality and differences in immune phenotypes in splenic or Peyer’s patch lymphocytes; these differences in immune phenotypes could be rescued through co-housing mice with wild-type animals, which results in improved T cell responses against bacterial antigens and directed IgA production, showing the interplay between the microbiome and the host immune response to sepsis [18,36,37]. While these microbial disturbances in sepsis patients are incompletely understood, they are thought to result from sepsis itself and managing interventions, as factors such as hypoxic injury and inflammation, intestinal dysmotility, shifts in intraluminal pH values, treatment with vasopressors, proton-pump inhibitors, opioids, and parenteral or enteral feeding can all influence the microbiome [6,38]. 

The gut microbiota in turn has emerged as a potential predictor of clinical outcomes in sepsis [6,18,19,20,36]. Gut microbial homeostasis prevents colonization with multi-drug-resistant bacteria; therefore, it has been posited that shifts in microbial composition may predispose to immunosuppression and increased sepsis risk [36]. There has been supporting evidence from preclinical studies using Western-diet mice. For example, after antibiotic treatment and an otherwise recoverable sterile surgical injury, mice fed an obesogenic Western diet were more susceptible to lethal sepsis [36,39]. This was attributed to a loss of *Bacteroidetes* and increased *Proteobacteria* [36,39]. Yet, others have argued that sepsis severity and mortality may be independent of the microbiome, but rather that other mechanisms leading to dysregulation of the innate immune system may play a larger role, though these mechanisms have not been clarified [40].

Clinical studies have been more variable in reporting microbial changes in septic patients, though general trends exist. For instance, several studies have reported that surgical patients with antibiotic-induced dysbiosis had a significantly increased risk of sepsis, and that this dysbiosis puts patients at an increased risk for a subsequent hospitalization for sepsis [41,42,43]. Other studies have tried to clarify the role of specific phyla. Intestinal domination of *Enterococcus* (>30% relative abundance) or colonization with vancomycin-resistant *Enterococcus* has been associated with a 19% risk for death in critically ill ICU patients [44]. In particular, septic patients who died were found to have differentially increased abundance of *Enterococcus* [44]. Others have validated this finding of differentially expressed *Enterococcus* with some reporting that an increase of one logarithmic unit in the abundance of certain *Enterococcus* phylotypes leads to a 3.14-fold increase in the probability of death in the ICU by sepsis [19,34,45]. *Parabacteroides distasonis* and *Bilophila* spp. have also been reported to be phylotypes exhibiting the most extreme association with sepsis samples. In line, *P. distasonis* plays key roles in surface antigen-mediated attenuation of the immune response to enhance endotoxin; these bacteria have further been implicated in antibiotic resistance and, accordingly, with increased mortality risk in critically ill patients [34,46]. The observed increases in *Bilophila* spp. together with *Fusobacterium* species in sepsis samples are strongly associated with the onset and progression of colorectal cancer; this further has implicated alterations in the gut bacteria with dysfunction of the entire organ, predisposing to disease [34,46].

Other clinical studies have further contributed that the *Firmicutes/Bacteroidetes* (F/B) ratio may be an important prognostic indicator [21]. In a study of intensive care patients, it was shown that critically ill patients with an F/B ratio of <0.1 or >10 had poorer prognoses, with survivors not falling into these indexes [47]. However, other studies have indicated no such effect [31,48]. In a randomized controlled trial of the impact of pre-treatment with broad-spectrum antibiotics on outcomes in healthy young men given intravenous LPS, some have reported no effects on surrogate markers of sepsis severity, including vital signs and fibrinolysis, despite decreased α-diversity and lower abundance of several beneficial gut bacteria [30]. While there are inconsistencies, in sum, all of this evidence hints that altering the composition of the gut microbiome may affect sepsis risk and outcomes, though gut microbiome composition and diversity are unlikely to account for all the clinical heterogeneity in sepsis [18].

## 3. Interactions between Opioid Use and Sepsis: The Role of the Microbiome and microRNAs

### 3.1. Opioids Increase the Risk of Sepsis in Clinical and Preclinical Models

Due to their analgesic properties, opioids are widely used in ICU settings for pain management in critically ill septic patients to optimize patient comfort and facilitate mechanical ventilation [49,50,51]. Furthermore, post-operative infections are a leading cause of sepsis in hospitalized patients, who are often maintained on opioids for pain management. Despite the frequent use of opioids in ICU settings, several studies have demonstrated an association between opioid use and increased risk of sepsis.

In a retrospective cohort study using electronic health records, it was found that opioid-treated patients with sepsis had substantially increased mortality rates compared to non-opioid-treated patients [50]. After adjusting for demographics, clinical comorbidities, severity of illness, and type of infection, opioid-treated patients had a significantly increased mortality rate at 28 days [50]. Laboratory cultures revealed that the most prevalent Gram-positive bacteria were *Staphylococcus*, *Streptococcus*, and *Enterococcus*, and the most prevalent Gram-negative bacteria were *E. coli*, *Salmonella*, and *Campylobacter*, consistent with other reports of commonly implicated bacteria in sepsis [50]. Similarly, others have found dramatically increased levels of endogenous morphine in the serum of patients with generalized infection in sepsis, severe sepsis (sepsis + end organ damage, hypotension, elevated lactate levels), and septic shock (severe sepsis + refractory hypotension) compared with inflammatory states without infection such as systemic inflammatory response syndrome (SIRS) (Temp. ≥ 38 °C or ≤36 °C; HR ≥ 90; RR ≥ 20 or PaCO_2_ < 32; WBCs ≥ 12,000 or ≤4000 or ≥10% bands) [52].

In line, several murine sepsis models have demonstrated that opioids promote sepsis progression and increase risk of opportunistic infections [53,54,55,56,57,58,59]. Using CLP, a well-established model for inducing poly-microbial sepsis, it was demonstrated that both morphine and methadone treatment resulted in high mortality following CLP when compared with placebo-treated animals [53]. Furthermore, it was found that LPS stimulation following morphine treatment increased proinflammatory cytokines TNF-α, IL-1β, and IL-6, and reduced serum corticosterone, which act synergistically to enhance the development of sepsis and septic shock [55]. Morphine treatment has consistently been shown to spontaneously induce sepsis in mice and promote bacterial translocation to the liver, spleen, and peritoneal cavity; additionally, morphine treatment also confers hyper-susceptibility to sublethal endotoxin challenges [54,56,60,61,62]. Further, morphine treatment has been shown to significantly potentiate immunosuppression in septic-animal models [54,56]. Interestingly, opioid antagonism with naloxone or naltrexone can block acute septic shock, providing further evidence of the profound role of opioids in the progression of sepsis [63]. While the exact mechanisms by which opioids modulate sepsis progression is still an area of active research, several studies have implicated opioid induction of microbial dysbiosis and immune system modulation as potential mechanisms.

### 3.2. Opioids Induce Gut Microbial Dysbiosis, Which Is Associated with Increased Sepsis Risk

Accumulating studies have implicated opioid-induced microbial dysbiosis in the development of sepsis and septic shock. It is well-established that opioids dysregulate the balance between intestinal epithelial cells (IECs), host immunity, and the microbiome [53,60,61,64].

We and others have previously shown that chronic opioid treatment alters gut microbial composition and induces preferential expansion of Gram-positive pathogenic strains [53,60,61,64]. In particular, using a murine model of polymicrobial sepsis, morphine treatment resulted in enrichment of the Firmicutes phylum, specifically the Gram-positive bacteria *Staphylococcus* and *Enterococcus* in the gut lumen [53]; furthermore, these same species were found to translocate to other systemic organs. Clinical studies have consistently reported that *S. aureus* is a commonly isolated strain from patients with Gram-positive sepsis; similarly, infection with *Enterococcus* is known to be an independent factor associated with a greater risk of death amongst hospitalized patients [32,53].

Intestinal bacterial translocation following opioid-induced sepsis is attributed to damaged tight junctions between IECs [60]. Consistently, intestinal sections from sepsis patients on opioids have shown disruption in gut epithelial integrity [53]. In murine models, protective microbial metabolites (such as short-chain fatty acids) normally responsible for maintaining the gut barrier were found to be decreased in morphine-treated groups, resulting in increased permeability of the gut epithelium and enhanced bacterial translocation [53,60,61,64]. This finding has since been validated in multiple other clinical and preclinical studies using microbial metabolites such as short-chain fatty acids and butyrate to improve GI epithelial barrier integrity.

Toll-like Receptor 4 (TLR4) knockout animals have further shed light into the underlying mechanisms of opioid-induced tight junction damage and bacterial translocation. These knockouts have largely attenuated opioid-mediated gut barrier damage [60]. In addition, TLR2 and 4 are reported to be upregulated in opioid-exposed mice, leading to overexpression of IL-17A and IL-6 [53,60]. Overexpression experiments have further shown that IL-17A is involved in destabilizing the gut epithelium, as IL-17A overexpression led to impaired intestinal epithelial barrier function, sustained bacterial dissemination, and elevated systemic inflammation [53]. Conversely, IL-17A neutralization was demonstrated to protect barrier integrity and improve survival in morphine-treated animals.

The commensal pool of the gut microbiome is a common origin of sepsis-inducing bacteria [56,60]. As such, these data indicate that opioid-induced changes in the composition or density of the microbiota may predispose to opportunistic pathogens [65].

### 3.3. Opioid Modulation of Immune Cells may Increase Sepsis Risk

Additionally, opioids are well known to modulate host immune function, which also may predispose to sepsis. While the systemic response to chronic opioid use is tolerance, this is not true for immune cells, which continue to be influenced by opioid use [66]. The immunomodulatory response to opioids in patients with prolonged opioid use or a history of opioid abuse has been extensively described [59]. These studies have shown that chronic morphine significantly impairs innate immunity, including a decrease in cytokine and chemokine production, macrophage phagocytosis, neutrophil migration, natural killer cell cytotoxicity, and dendritic cell antigen presentation, amongst several other effects on immune cells [57,59,67]. Additionally, opioids have profound effects on adaptive immunity, resulting in decreased TH1 cytokine production, T cell activation, and B cell proliferation, antibody production, and MHC-II expression [59,67]. This opioid-mediated immune dysregulation leads to decreased pathogen clearance, which increases susceptibility to opportunistic infections, commonly observed in the opioid-using population [57,59]. Despite our vast knowledge of how opioids affect the innate and adaptive arms of the immune system, the molecular mechanisms by which morphine modulates immunity and increases susceptibility to bacterial infection are only beginning to be understood.

Recently, there has been increasing focus on opioid modulation of microRNAs (miRNAs) in innate immune cells involved in the regulation of the inflammatory response by microbiota-derived antigens. Pathogens have been shown to epigenetically regulate gene expression in innate immune cells to control the inflammatory response, which contributes to the morbidity and the mortality of sepsis [68,69]. miRNAs are small non-coding RNAs which post-transcriptionally mediate protein expression by targeting specific mRNA degradation [68,70]. Multiple miRNAs have been shown to regulate the LPS-induced inflammatory cascade [71]; among them, miR-146a has emerged as one of the most important miRNAs orchestrating immune and inflammatory signaling [72,73]. miR-146a plays a key role in immune tolerance. Under LPS stimulation, miR-146a is positively regulated by NF-κB p65; however, miR-146a also acts in a negative feedback loop to downregulate NF-κB p65 by targeting MyD88 and TRIF degradation, which promotes immune tolerance [74]. Thus, miR-146a is a negative effector of the innate immune response, preventing overstimulation of the inflammatory response to bacterial endotoxin [75]. This has further been confirmed with miR-146a knockout mice, which are hyperresponsive to LPS stimulation [75].

Interestingly, opioids have been shown to modulate miRNAs (Table 1) which affect the establishment of endotoxin tolerance. In a murine model of chronic opioid usage, LPS induced less production of miR-146a and miR-155 in macrophages of morphine-treated animals compared to placebo, resulting in reduced endotoxin-induced tolerance response assessed by IL-6 levels [66]. This result was reversed in miR-146a knockdown animals, with targets of miR-146a and miR-155 (IRAK1, TRAF6, and TAB2, respectively) upregulated with chronic morphine exposure [66]. However, only miR-146a overexpression, but not miR-155, was reported to attenuate morphine-mediated hyper-inflammation [66]. Furthermore, it was demonstrated in a morphine-tolerance rat model that miR-146a was greatly decreased in the spinal cord [76]; lentiviral overexpression of miR-146a failed to attenuate the development of morphine analgesic tolerance [76]. Still, mixed evidence exists related to opioid modulation of miR-146. A clinical study demonstrated an increase in plasma Let-7 family and miR-146a miRNAs after 24hrs of hydromorphone or oxycodone [71]. Let7 miRNAs have a wide variety of effects [77], but there is evidence that Let7c miRNAs also can downregulate IL-6 production by binding to the IL6 3′ UTR, contributing to endotoxin tolerance [78]. Additionally, others have found that morphine induces an increase in miR-146a in human monocyte-derived macrophages, though there was <0.5-fold change in the expression level as compared to controls [79].

Recently, a study using intestinal organoids has shed insight into the morphine modulation of miRNAs through extracellular vesicles (EVs) in response to LPS [83]. EVs produced by intestinal epithelial cells deliver miRNAs and other signaling molecules to intestinal immune cells to coordinate homeostasis or the immune response [84,85]. Interestingly, many opioid-induced extravesicular miRNAs are associated with TLR signaling, specifically the Let-7 miRNA family and miR-146. While EVs from intestinal organoids without morphine treatment were shown to alleviate LPS-mediated inflammation in vitro, EVs from morphine-exposed organoids did not [83]. Specifically, the Let-7 family of miRNAs were downregulated in EVs produced by morphine-treated organoids [83]. While miR-146a levels have been shown to be altered by morphine treatment, in this study they were unchanged in the EVs, suggesting other mechanisms may be at play in modifying miR-146a expression [83].

Together, these clinical and laboratory data strongly suggest that opioids compromise the defensive functions of immune cells necessary for pathogen clearance and tapering of the inflammatory cascade; this in turn disturbs the homeostasis between the pro- and anti-inflammatory pathways in response to pathogens that can lead to sepsis (Figure 2).

## 4. MicroRNA Modulation of Sepsis

The observation that morphine-induced exacerbation of sepsis is mediated by tempering endotoxin tolerance through modulation of miR-146a [66] is particularly fascinating as miRNAs are widely implicated in sepsis [86,87].

miRNA have been recognized as critical regulators of endotoxin tolerance through TLR signaling. In particular, miR-146a has been thoroughly described to reduce the severity of sepsis in murine models (Table 2). miR-146a binds to critical components downstream of TLR4 signaling, such as IRAK1 and TRAF6, as well as other signaling pathways such as Notch-1, which lessen the inflammatory response [88]. For instance, LPS induces up-regulation of miR-146a in macrophages, which is protective against LPS-induced organ damage [88]. Similarly, during sepsis, the addition of miR-146a has been shown to attenuate hyper-inflammation and prevent multiple organ failure [9]. In vitro, miR-146a most profoundly suppressed the production of pro-inflammatory cytokines in RAW264.7 macrophage cells after LPS stimulation out of all exogenously applied miRNAs [9]. Injection of miR-146a has further been reported to prevent the development of cardiomyopathy in a rat sepsis model, again showing the important functions of miR-146a in the prevention of sepsis [89]. However, conflicting evidence exists; other studies have shown that miR-146a deficiency can be protective against *Listeria monocytogenes* infection, with significantly increased abundance of short-chain-fatty-acid-producing bacteria in miR-146a-deficient mice [90].

Given the assumed protective roles of miR-146a in sepsis progression, there has been an increasing focus on epigenetic regulators of miR-146a to control the excessive inflammatory response in microbial sepsis. For instance, IL-1β-primed mesenchymal stem cells have been shown to dampen hyper-inflammation in murine sepsis models [82]. Pre-stimulated mesenchymal stem cells have gained popularity because of their anti-inflammatory functions, as summarized in Wang 2014. Additionally, IL-1β-primed mesenchymal stem cells produce exosomes containing large amounts of miR-146a, which are thought to mediate the anti-inflammatory response [82]. Others have used pharmacological inhibition of JMJD3, which modulates miR-146a transcription, to protect mice against early septic death [68]. Specifically, it was found that inhibition of JMJD3 by GSKJ4, a small-molecule inhibitor, decreased the expression of inflammatory mediators [68]. JMJD3 was further shown to negatively regulate the transcription of miR-146a via its demethylation of H3K27me3 on the promoter of miR-146a, promoting an anti-inflammatory response [68].

Clinical studies have also reflected the observation that miR-146a may be protective against sepsis. Decreased plasma miR-146a has been reported with sepsis patients compared to non-sepsis SIRS patients [93]. Furthermore, mutations in the miR-146a gene are associated with an increased risk of sepsis [94]. Collectively, miR-146a appears to play an immunomodulatory role to reduce the pro-inflammatory response in both animal models and humans, highlighting its role as a potential sepsis biomarker.

## 5. Therapeutic Modulation of Sepsis

Recent research describing the role of the gut microbiome in sepsis has prompted the development of several microbiome-related preventative and therapeutic strategies. One approach for the eradication of potential pathogenic GI bacteria has been through selective digestive tract decontamination (SDD) using antimicrobials against resistant pathogens. However, SDD has raised concerns over re-colonization of resistant pathogens or selection of more resistant organisms after treatment [95,96]. To date, probiotics, prebiotics, and synbiotics remain the most studied and well-evaluated approaches to prevent sepsis and improve sepsis outcomes. Whereas probiotics are defined as living microorganisms with potential beneficial properties, prebiotics are a non-digestible food ingredient that stimulate the growth or activity of gut bacteria, with synbiotics being a combination of both probiotics and prebiotics. 

Several studies have shown select benefits in sepsis risk with microbiome modulation. In a mouse model of sepsis, Chen et al. illustrated that the mortality of septic mice was significantly lower with pre-administration of *Lactobacillus rhamnosus* GG [97]. A randomized, double-blind, placebo-controlled trial of 4556 healthy infants in rural India showed that an oral synbiotic preparation (*Lactobacillus* plantarum plus fructooligosaccharide) significantly reduced sepsis infection and death [98]. However, in a multicenter, randomized controlled phase 3 study of 1315 preterm infants at high risk of sepsis, the probiotic *Bifidobacterium breve* BBG-001 did not show preventative benefits in either necrotizing enterocolitis or late-onset sepsis [99]. These findings indicate that probiotics on their own may not be sufficient to prevent sepsis and demonstrate that preventative effects may be dependent on unique bacterial species, growth components, and metabolites in probiotic and synbiotic preparations. In addition to the widely studied bacteria *Lactobacillus* and *Bifidobacterium*, several next-generation probiotics have shown promising benefits. These include, but are not limited to, *Akkermansia*, which was shown to be associated with increased sepsis survival in mice [100]; a probiotic consortium, which showed decreased colonization by vancomycin-resistant *Enterococcus* in antibiotics-treated mice [92]; and Lantibiotics, which are produced by a specific *Blautia producta* strain and were found to reduce colonization by vancomycin-resistant *Enterococcus* [80].

Fecal microbiota transplantation (FMT) has also been investigated as a potential therapeutic in sepsis. FMT is the administration of fecal material from a healthy donor into the intestinal tract of a patient with an altered gut microbiome to restore normal gut function. Unlike probiotics and synbiotics, which constitute only one or a small number of bacterial species, FMT has the potential effect of colonizing the entire host gut microbiome into the recipient. Several cases of FMT for sepsis patients have been reported [101,102,103]. In most cases, FMT has resulted in successful reversal of gut dysbiosis and has improved sepsis outcomes. However, variability between studies highlight the need for cautious donor screening, careful identification of patients, evaluation of the best route of administration (colonoscopy or enema vs. nasogastric tract), and prudent decision making between autologous or heterologous transplantation to leverage the benefits of FMT in sepsis prevention and treatment.

As microRNAs modulate endotoxic tolerance to inflammation and have widely been implicated in sepsis, how microRNAs may further be impacted by probiotics is another area of great interest that will need to be thoroughly explored. miRNAs play key roles in host–microbiome interactions and can modulate gut microbiota. Ahmed et al. (2009), Link et al. (2012), and Liu et al. (2016) studied gut microRNAs in intestinal contents and feces and showed that microRNAs were able to affect gut microbiota composition [91,104,105]. Specifically, it was found that these host-derived microRNAs can enter bacterial cells and influence bacterial gene expression as well as growth [91]. For instance, miR-146a has been shown to regulate crosstalk between intestinal epithelial cells, microbial components, and inflammatory stimuli [72]. However, there have been few reports on how miRNAs, and in particular miR-146a, are impacted by probiotics. One study has reported differential induction of miR-146a by pathogenic and probiotic *E. coli* strains in epithelial and immune cells, further supporting miR-146a-gut microbiome interactions and the potential to target microRNA expression through probiotics [81]. Still, much work is needed in this area.

## 6. Conclusions

In summary, while sepsis is a leading cause of morbidity and mortality in hospitalized patients, the mainstays of treatment (antibiotic and supportive care) have remained unchanged for decades. Thus, there has been extreme urgency in understanding immune dysregulation in sepsis and contributing mechanisms. Opioids are commonly prescribed in ICU settings for pain management; additionally, the growing incidence of opioid use disorders increases the risk for infection-related hospitalizations. Thus, understanding how opioids affect sepsis risk is of paramount importance. Overall, clinical and preclinical studies support the notion that opioids increase sepsis risk and may do so through microbiome and immune modulation. The finding that miR-146a, which plays key roles in endotoxic tolerance, is also modulated by opioids to prevent immune tolerance following inflammation also provides possible mechanistic insight as to how opioids may increase sepsis risk. As both opioid use and predisposition to sepsis are associated with microbial dysbiosis, several studies have tried to modulate the microbiome to improve sepsis risk in opioid-using patients, with some successful attempts. While not much information exists regarding pharmacological modulation of miRNAs, particularly miR-146a, there is some evidence of protective effects that necessitate further exploration.

## Figures and Tables

**Figure 1 ijms-23-01097-f001:**
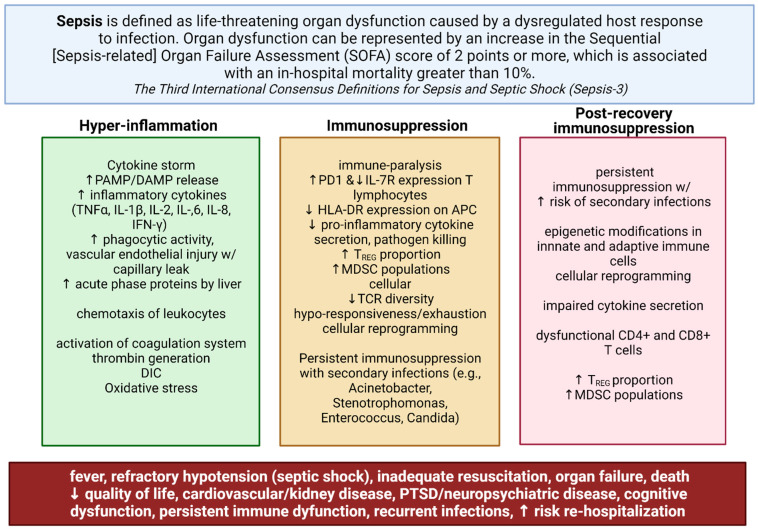
Dysregulated immune responses in sepsis. Sepsis has recently been defined as life-threatening organ dysfunction caused by the dysregulated host response to an ongoing or suspected infection. Sepsis is marked by simultaneous inflammation and immune suppression, which may persist well past the immediate recovery period. The excessive response of the innate immune system following bacterial invasion results in an overwhelming inflammatory response known as a cytokine storm. Later immunosuppression during sepsis contributes to increased 6- and 12-month mortality rates in up to 20% of survivors of sepsis and septic shock primarily from secondary infections; this results in organ dysfunction and failure. Supportive care and broad-spectrum antibiotic treatment—the mainstays of treatment—have not significantly improved morbidity and mortality amongst hospitalized patients, or patients in the post-recovery period due to persistent immunosuppression and immune dysfunction. Created with BioRender.com.

**Figure 2 ijms-23-01097-f002:**
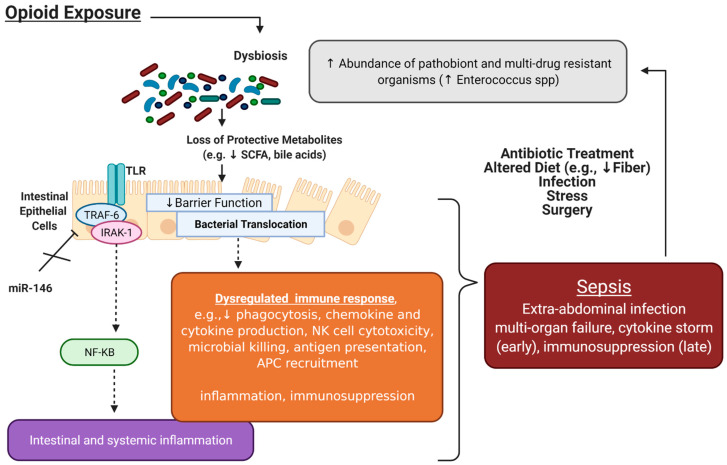
Interactions between opioids, the microbiome, and sepsis. Accumulating studies have implicated opioid-induced microbial dysbiosis in the development of sepsis and septic shock. The microbiome plays a critical role in intestinal homeostasis; under physiological conditions, intestinal microbial homeostasis promotes resistance to pathogen colonization through maintenance of the gut epithelial barrier, production of short-chain fatty acids (SCFA), and regulation of immune cells. Chronic opioid treatment results in loss of gut commensals and protective metabolites responsible for maintaining the gut epithelial barrier. Increased intestinal epithelial cell permeability allows for bacterial translocation in extra-intestinal spaces, which results in intestinal and systemic inflammation (mediated by TLR signaling) and dysregulated immune responses. Mir-146a, which plays a key role in immune tolerance by inhibiting IRAK1 and TRAF6, as well as other signaling pathways such as Notch-1 to lessen the inflammatory response, has been shown to be attenuated with morphine treatment. Altogether, this dysregulated immune response can predispose to sepsis. The mainstays of sepsis treatment, in particular antibiotics, further contribute to gut microbial dysbiosis, perpetuating this cycle. Figure created with BioRender.com.

**Table 1 ijms-23-01097-t001:** Summary of literature describing opioid modulation of miR-146a. Opioids have been extensively described to modulate miR-146a in clinical and pre-clinical models.

Opioid Modulation of miR-146a		
**Model**	Key Findings	Reference
Murine	morphine ↓ endotoxin/LPS induced miR-146a and 155 expression in macrophages; only miR-146a overexpression, not miR-155, abrogates morphine-mediated hyper-inflammation; antagonizing miR-146a ↑ severity of morphine-mediated hyper-inflammation	[11]
Murine	↓ miR-146a in spinal cord in a morphine-tolerance rat model; lentiviral overexpression of miR-146a fails to attenuate the development of morphine analgesic tolerance	[80]
Murine	miR-146a levels unchanged in the extracellular vesicles after morphine treatment	[81]
Human	↑ plasma Let-7 family and miR-146a expression after 24 h of hydromorphone or oxycodone treatment	[82]
Human	morphine ↑ miR-146a expression in human monocyte-derived macrophages	[19]

**Table 2 ijms-23-01097-t002:** Summary of literature describing role of miR-146a in sepsis and SIRS models. miR-146a is a critical regulator of endotoxin tolerance and has been shown to affect sepsis development in clinical and preclinical models.

Role of miR-146a in Sepsis and SIRS Models		
**Model**	Key Findings	Reference
Murine	↑ miR-146a in macrophages protective against LPS-induced organ damage	[10]
Murine	miR-146a ↓ hyper-inflammation and prevents multiple organ failure in sepsis; miR-146a ↓ production of pro-inflammatory cytokines in RAW264.7 macrophage cells after LPS stimulation	[28]
Murine	miR-146a ↓ development of sepsis-induced cardiomyopathy by regulating TLR4/NF-κβ signaling pathway	[91]
Murine	miR-146a-deficient mice more resistant to *L. monocytogenes* infection via modulation of the gut microbiota	[22]
Murine	IL-1β-primed mesenchymal stem cells, which produce miR-146a, ↓ hyper-inflammation in murine sepsis models	[86]
Murine	inhibition of JMJD3, which modulates miR-146a transcription, protects mice against early septic death	[65]
Human	↓ plasma miR-146a in sepsis patients compared to non-sepsis SIRS patients	[92]
Human	mutations in the miR-146a gene associated with an ↑ risk of sepsis	[84]

## Data Availability

Not applicable.

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
