# Peer review of "Opioids and Sepsis: Elucidating the Role of the Microbiome and microRNA-146"

_ijms, 2022, doi:10.3390/ijms23031097_

Round 1

Reviewer 1 Report

Look at these notes:

  • It is not clear if this is a review or an article research. Probably a review, but this needs to be stated, and then the paper accordingly set. 
  • Please add a Methods section 
  • It is not clear in the text introduction what is the aim of this paper. Please improve. 
  • Some recent refs should be considered: Oral Microbiome and Preterm Birth: Correlation or Coincidence? A Narrative Review. Open Access Maced. J. Med. Sci. 2020, 8, 123–132.  DOI: 10.3889/oamjms.2020.4444  ----  From focal sepsis to periodontal medicine: a century of exploring the role of the oral microbiome in systemic disease. J Physiol. 2017 Jan 15;595(2):465-476. doi: 10.1113/JP2724 --- Preterm Birth Is Correlated With Increased Oral Originated Microbiome in the Gut. Front Cell Infect Microbiol. 2021 Jun 17;11:579766. doi: 10.3389/fcimb.2021.579766.
  • Table 1 needs some explanation in the table legend

Author Response

Thank you for your thoughtful comments for our review entitled: Opioids and sepsis: elucidating the role of the microbiome and microRNAs. Below, we address the Reviewer’s comments in a point-by-point-manner.

Reviewer 1:

  • It is not clear if this is a review or an article research. Probably a review, but this needs to be stated, and then the paper accordingly set.
    • We have clarified that this is a review.
    • See page 4 paragraph 4- “In this review, we will describe the role of the gut microbiota in sepsis, and interactions between sepsis and opioids.”
  • Please add a Methods section
    • We have clarified that this is a review, and not an original article.
  • It is not clear in the text introduction what is the aim of this paper. Please improve.
    • We have included the aims of this review – See page 4 paragraph 4.
  • Some recent refs should be considered:
    • While interesting reviews, our own review is focused specifically on the gut microbiome, and not the oral microbiome. Preterm birth may also be out of the scope of this review, though an interesting field.
  • Table 1 needs some explanation in the table legend
    • We have added a description for Tables.

Reviewer 2 Report

The authors presented in this review manuscript regarding the contributing roles of microRNA, especially miR-146, in modulating sepsis pathogenesis and therapeutic targeting of the gut microbiome in controlling immune dysregulation in sepsis. I think this manuscript is sufficient to provide the new insight about the application of modulating sepsis pathogenesis through the regulation of miR expression and its activity. However, minor revision is needed to be published as below. 

1) The title of manuscript is "Opioids and sepsis: elucidating the role of the microbiome and microRNAs", but it is mainly focused at miR-146. Considering that this is a review paper, there should also be a description of miRNAs other than miR-146.

2) The legend of Table 1 should be more described as detail as possible. 

Author Response

Thank you for your thoughtful comments for our review entitled: Opioids and sepsis: elucidating the role of the microbiome and microRNAs. Below, we address the Reviewer’s comments in a point-by-point-manner.

Reviewer 2:

1) The title of the manuscript is "Opioids and sepsis: elucidating the role of the microbiome and microRNAs", but it is mainly focused at miR-146. Considering that this is a review paper, there should also be a description of miRNAs other than miR-146.

  • We have changed the title of the paper to reflect our focus on miR-146.
  • Of the microRNA’s, Mir-146 has been most extensively described to play a role in opioids, sepsis, and the microbiome; a focus on this microRNA provides enough literature for a comprehensive review, as opposed to the other microRNAs that are understudied with regards to both the microbiome and sepsis.

2) The legend of Table 1 should be more described as detail as possible.

  • We have added a description for Tables.

Round 2

Reviewer 1 Report

Well done